# VERIFICATION OF NON-LINEAR SPECIFICATIONS FOR NEURAL NETWORKS

**Chongli Qin**[*]**, Krishnamurthy (Dj) Dvijotham**[*]**, Brendan O'Donoghue, Rudy Bunel, Robert Stanforth, Sven Gowal, Jonathan Uesato, Grzegorz Swirszcz, Pushmeet Kohli**

DeepMind
London, N1C 4AG, UK
correspondence: chongliqin@google.com

## ABSTRACT

Prior work on neural network verification has focused on specifications that are *linear* functions of the output of the network, e.g., invariance of the classifier output under adversarial perturbations of the input. In this paper, we extend verification algorithms to be able to certify richer properties of neural networks. To do this we introduce the class of *convex-relaxable specifications*, which constitute nonlinear specifications that can be verified using a convex relaxation. We show that a number of important properties of interest can be modeled within this class, including conservation of energy in a learned dynamics model of a physical system; semantic consistency of a classifier's output labels under adversarial perturbations and bounding errors in a system that predicts the summation of handwritten digits. Our experimental evaluation shows that our method is able to effectively verify these specifications. Moreover, our evaluation exposes the failure modes in models which cannot be verified to satisfy these specifications. Thus, emphasizing the importance of training models not just to fit training data but also to be consistent with specifications.

## 1 INTRODUCTION

Deep learning has been shown to be effective for problems in a wide variety of different fields from computer vision to machine translation (see Goodfellow et al. (2016); Sutskever et al. (2014); Krizhevsky et al. (2012)). However, due to the black-box nature of deep neural networks, they are susceptible to undesirable behaviors that are difficult to diagnose. An instance of this was demonstrated by Szegedy et al. (2013), who showed that neural networks for image classification produced incorrect results on inputs which were modified by small, but carefully chosen, perturbations (known as *adversarial perturbations*). This has motivated a flurry of research activity on designing both stronger attack and defense methods for neural networks (Goodfellow et al., 2014; Carlini & Wagner, 2017b; Papernot et al., 2016; Kurakin et al., 2016; Athalye et al., 2018; Moosavi-Dezfooli et al., 2016; Madry et al., 2017). However, it has been observed by Carlini & Wagner (2017b) and Uesato et al. (2018) that measuring robustness of networks accurately requires careful design of strong attacks, which is a difficult task in general; failure to do so can result in dangerous underestimation of network vulnerabilities.

This has driven the need for *formal verification*: a provable guarantee that neural networks are consistent with a *specification* for all possible inputs to the network. Remarkable progress has been made on the verification of neural networks (Tjeng & Tedrake, 2017; Cheng et al., 2017; Huang et al., 2017; Ehlers, 2017; Katz et al., 2017; Bunel et al., 2017; Weng et al., 2018; Gehr et al., 2018; Wong & Kolter, 2018; Dvijotham et al., 2018b). However, verification has remained mainly limited

---

[*]equal contribution

to the class of *invariance specifications*, which require that the output of the network is invariant to a class of transformations, e.g., norm-bounded perturbations of the input.

There are many specifications of interest beyond invariance specifications. Predictions of ML models are mostly used not in a standalone manner but as part of a larger pipeline. For example, in an ML based vision system supporting a self-driving car, the predictions of the ML model are fed into the controller that drives the car. In this context, what matters is not the immediate prediction errors of the model, but rather the consequences of incorrect predictions on the final driving behavior produced. More concretely, it may be acceptable if the vision system mistakes a dog for a cat, but not if it mistakes a tree for another car. More generally, specifications that depend on the *semantics of the labels* are required to capture the true constraint that needs to be imposed.

**Our Contributions.** Invariance specifications are typically stated as linear constraints on the outputs produced by the network. For example, the property that the difference between the logit for the true label and any incorrect label is larger than zero. In order to support more complex specifications like the ones that capture label semantics, it is necessary to have verification algorithms that support nonlinear functions of the inputs and outputs. In this paper, we extend neural network verification algorithms to handle a broader class of *convex-relaxable specifications*. We demonstrate that, for this broader class of specifications verification can be done by solving a convex optimization problem. The three example specifications that we study in this paper are: a) Physics specifications: Learned models of physical systems should be consistent with the laws of physics (e.g., conservation of energy), b) Downstream task specifications: A system that uses image classifiers on handwritten digits to predict the sum of these digits should make only small errors on the final sum under adversarial perturbations of the input and c) Semantic specifications: The expected semantic change between the label predicted by a neural network and that of the true label should be small under adversarial perturbations of the input. Finally, we show the effectiveness of our approach in verifying these novel specifications and their application as a diagnostic tool for investigating failure modes of ML models.

**Related work.** Verification of neural networks has received significant attention in both the formal verification and machine learning communities recently. The approaches developed can be broadly classified into *complete* and *incomplete* verification. Complete verification approaches are guaranteed to either find a proof that the specification is true or find a counterexample proving that the specification is untrue. However, they may require exhaustive search in the worst case, and have not been successfully scaled to networks with more than a few thousand parameters to-date. Furthermore, these exhaustive approaches have only been applied to networks with piecewise linear activation functions. Examples of complete verification approaches include those based on Satisfiability Modulo Theory (SMT) solvers (Huang et al., 2017; Ehlers, 2017; Katz et al., 2017) and those based on mixed-integer programming (Bunel et al., 2017; Tjeng & Tedrake, 2017; Cheng et al., 2017). Incomplete verification algorithms, on the other hand, may not find a proof even if the specification is true. However, if they do a find a proof, the specification is guaranteed to be true. These algorithms are significantly more scalable thus can be applied to networks with several hundred thousand parameters. Moreover, they have been extended to apply to arbitrary feedforward neural networks and most commonly used activation functions (sigmoid, tanh, relu, maxpool, resnets, convolutions, etc.). Incomplete verification algorithms include those based on propagating bounds on activation functions (Weng et al., 2018; Mirman et al., 2018a) and convex optimization (Dvijotham et al., 2018b). By making verification both scalable to larger models and more efficient – these approaches have enabled verification to be folded into training so that networks can be trained to be provably consistent with specifications like adversarial robustness (Mirman et al., 2018b; Wong et al., 2018; Raghunathan et al., 2018).

**Organization of the Paper** In Section 2 we give examples of different non-linear specifications which neural networks should satisfy, in Section 3 we formalize our approach and use it to show how the non-linear specifications in Section 2 can be verified. Finally, in Section 4 we demonstrate experimental results.

## 2    Specifications beyond robustness

In this section, we present several examples of verification of neural networks beyond the usual case of robustness under adversarial perturbations of the input – all of which are non-linear specifications.

There has been research done on relaxations of non-linear activation functions including ReLU (Wong & Kolter, 2018; Dvijotham et al., 2018a; Weng et al., 2018; Mirman et al., 2018a), sigmoid, tanh and other arbitrary activation functions (Dvijotham et al., 2018b; Huang et al., 2017). However, there is a distinction between relaxing non-linear activations vs non-linear specifications. The former assumes that the relaxations are done on each neuron independently, whereas in non-linear specifications the relaxations will involve interactions across neurons within the neural network such as the softmax function.

Prior verification algorithms that focus on adversarial robustness are unable to handle such nonlinearities, motivating the need for the new algorithms developed in this paper.

**Specifications derived from label semantics:**    Supervised learning problems often have semantic structure in the labels. For example, the CIFAR-100 dataset, a standard benchmark for evaluating image classification, has a hierarchical organization of labels, e.g., the categories 'hamster' and 'mouse' belong to the super-category 'small mammals'. In many real world applications, misclassification within the same super-category is more acceptable than across super-categories. We formalize this idea by using a distance function on labels $d(i, j)$ – e.g., defined via the above mentioned hierarchy – and require the probability of a classifier making a 'large distance' mistake to be small. Formally, if we consider an input example $x$ with true label $i$, the expected semantic distance can be defined as $\mathbb{E}\left[d(i, j)\right] = \sum_j d(i, j)P(j|x)$, where $P(j|x)$ is the probability that the classifier assigns label $j$ to input $x$, and we assume that $d(i, i) = 0$. We can write down the specification that the expected distance is not greater than a threshold $\epsilon$ as

$$\mathbb{E}\left[d(i, j)\right] \leq \epsilon. \tag{1}$$

Here, we consider neural network classifiers where $P(j|x)$ is the softmax over the logits produced by the network. We note that this is a non-separable nonlinear function of the logits.

**Specifications derived from physics:**    An interesting example of verification appears when employing ML models to simulate physical systems (see Stewart & Ermon (2017); Ehrhardt et al. (2017); Bar-Sinai et al. (2018)). In this context, it may be important to verify that the learned dynamic models are consistent with the laws of physics. For example, consider a simple pendulum with damping, which is a dissipative physical system that loses energy. The state of the system is given by $(w, h, \omega)$ where $w$ and $h$ represent the horizontal and vertical coordinates of the pendulum respectively, and $\omega$ refers to the angular velocity. For a pendulum of length $l$ and mass $m$ the energy of the system is given by

$$E\left(w, h, \omega\right) = \underbrace{mgh}_{\text{potential}} + \underbrace{\frac{1}{2}ml^2\omega^2}_{\text{kinetic}}, \tag{2}$$

here $g$ is the gravitational acceleration constant. Suppose we train a neural network, denoted $f$, to model the next-state of the system given the current state: $(w', h', \omega') = f\left(w, h, \omega\right)$, where $(x', h', \omega')$ represents the state at the next time step (assuming discretized time). For this dissipative physical system the inequality

$$E\left(w', h', \omega'\right) \leq E\left(w, h, \omega\right) \tag{3}$$

should be true for all input states $\{w, h, \omega\}$. We note, this requires our verification algorithm to handle quadratic specifications.

**Specifications derived from downstream tasks:**    Machine learning models are commonly used as a component in a pipeline where its prediction is needed for a downstream task. Consider an accounting system that uses a computer vision model to read handwritten transactions and subsequently adds up the predicted amounts to obtain the total money inflow/outflow. In this system, the ultimate quantity of interest is the summation of digits - even if the intermediate predictions are incorrect, the system cares about the errors accumulated in the final sum. Formally, given a set of

handwritten transactions $\{x_n\}_{n=1}^N$ and corresponding true transaction values $\{i_n\}_{n=1}^N$, the expected error is

$$\mathbb{E}_j\left[|\sum_{n=1}^N(j_n-i_n)|\right] = \sum_{j\in J^N}|\sum_{n=1}^N(j_n-i_n)|\prod_{n=1}^N P(j_n|x_n),\tag{4}$$

where $J$ is the set of possible output labels. We would like to verify that the expected error accumulated for the inflow should be less than some $\epsilon > 0$

$$\mathbb{E}_j\left[|\sum_{n=1}^N(j_n-i_n)|\right] \leq \epsilon.\tag{5}$$

In contrast to the standard invariance specification, this specification cares about the error in the sum of predictions from using the ML model $N$ times.

## 3 FORMULATION OF NONLINEAR SPECIFICATIONS

We now formalize the problem of verifying nonlinear specifications of neural networks. Consider a network mapping $f : \mathcal{X} \to \mathcal{Y}$, with specification $F : \mathcal{X} \times \mathcal{Y} \to \mathbb{R}$ which is dependent on both the network inputs $x$ and outputs $y = f(x)$. Given $\mathcal{S}_{\text{in}} \subseteq \mathcal{X}$, the set of inputs of interest, we say that the specification $F$ is satisfied if

$$F(x,y) \leq 0 \quad \forall x \in \mathcal{S}_{\text{in}},\ y = f(x).\tag{6}$$

For example, the case of perturbations of input $x^{\text{nom}}$ in an $l_\infty$-norm ball uses

$$\mathcal{S}_{\text{in}} = \{x :\| x - x^{\text{nom}} \|_\infty \leq \delta\},\tag{7}$$

where $\delta$ is the perturbation radius. For this kind of specification, one way to solve the verification problem is by *falsification*, i.e., searching for an $x$ that violates (6) using gradient based methods (assuming differentiability). More concretely, we can attempt to solve the optimization problem

$$\max_{x\in\mathcal{S}_{\text{in}}} F(x, f(x))$$

using, for example, a projected gradient method of the form

$$x^{(i+1)} \leftarrow \text{Proj}\left(x^{(i)} + \eta_i\frac{dF(x,f(x))}{dx}, \mathcal{S}_{\text{in}}\right),\tag{8}$$

where $\eta_i$ is the learning rate at time-step $i$ and Proj denotes the Euclidean projection

$$\text{Proj}(x, \mathcal{S}_{\text{in}}) = \operatorname*{argmin}_{\zeta\in\mathcal{S}_{\text{in}}} \|x-\zeta\|.$$

If this iteration finds a point $x \in \mathcal{S}_{\text{in}}$ such that $F(x, f(x)) > 0$, then the specification is not satisfied by the network $f$. This is a well-known technique for finding adversarial examples (Carlini & Wagner, 2017a), but we extend the use here to nonlinear specifications. Maximizing the objective Eq. (6) for an arbitrary non-linear specification is hard. Since the objective is non-convex, gradient based methods may fail to find the global optimum of $F(x, f(x))$ over $x \in \mathcal{S}_{\text{in}}$. Consequently, the specification can be perceived as satisfied even if there exists $x \in \mathcal{S}_{\text{in}}$ that can falsify the specification, the repercussions of which might be severe. This calls for algorithms that can find *provable* guarantees that there is no $x \in \mathcal{S}_{\text{in}}$ such that $F(x, f(x)) > 0$ - we refer to such algorithms as *verification algorithms*. We note that, even when $F$ is linear, solving the verification problem for an arbitrary neural network $f$ has been shown to be NP-hard (Weng et al., 2018) and hence algorithms that exactly decide whether (6) holds are likely to be computationally intractable.

Instead, we settle for *incomplete verification*: i.e., *scalable* verification algorithms which can guarantee that the specification is truly satisfied. Incomplete algorithms may fail to find a proof that the specification is satisfied even if the specification is indeed satisfied; however, in return we gain computational tractability. In order to have tractable verification algorithms for nonlinear specifications, we define the class of *convex-relaxable specifications* - which we outline in detail in Section 3.2. We show that this class of specifications contains complex, nonlinear, verification problems and can be solved via a convex optimization problem of size proportional to the size of the underlying neural network. In later sections, we show how the specifications derived in the prior sections can be modeled as instances of this class.

## 3.1 Neural network

Here let $x \in \mathbb{R}^n, y \in \mathbb{R}^m$ denote either a single input output pair, i.e., $y = f(x)$ or multiple input output pairs (for multiple inputs we apply the mapping to each input separately). For classification problems, the outputs of the network are assumed to be the logits produced by the model (prior to the application of a softmax). For clarity of presentation, we will assume that the network is a feedforward network formed by $K$ layers of the form

$$x_{k+1} = g_k \left( W_k x_k + b_k \right), k = 0, \ldots, K - 1$$

where $x_0 = x$, $x_K = y$, and $g_k$ is the activation function applied onto the $k$th layer. While our framework can handle wider classes of feedforward networks, such as ResNets (see Wong et al. (2018)), we assume the structure above as this can capture commonly used convolutional and fully connected layers. Assume that the inputs to the network lie in a bounded range $\mathcal{S}_{\text{in}} = \{x : l_0 \le x \le u_0\}$. This is true in most applications of neural networks, especially after rescaling. We further assume that given that the input to the network is bounded by $[l_0, u_0]$ we can bound the intermediate layers of the network, where the $k$th hidden layer is bounded by $[l_k, u_k]$. Thus, we can bound the outputs of our network

$$\mathcal{S}_{\text{out}} = \{y : l_K \le y \le u_K\}. \tag{9}$$

Several techniques have been proposed to find the tightest bounds for the intermediate layers such as Wong & Kolter (2018); Bunel et al. (2017); Ehlers (2017); Dvijotham et al. (2018b). We refer to these techniques as bound propagation techniques.

## 3.2 Convex-relaxable specifications

Most prior work on neural network verification has assumed specifications of the form:

$$F(x, y) = c^T y + d \le 0 \quad \forall x \in \mathcal{S}_{\text{in}}, y = f(x) \tag{10}$$

where $c, d$ are fixed parameters that define the specification and $\mathcal{S}_{\text{in}}$ is some set. Here, we introduce the class of *convex-relaxable* specifications.

**Assumption 1.** *We assume that $\mathcal{S}_{\text{in}} \subseteq \mathbb{R}^n, \mathcal{S}_{\text{out}} \subseteq \mathbb{R}^m$ are compact sets and that we have access to an efficiently computable[1] procedure that takes in $F, \mathcal{S}_{\text{in}}, \mathcal{S}_{\text{out}}$ and produces a compact convex set $\mathcal{C}(F, \mathcal{S}_{\text{in}}, \mathcal{S}_{\text{out}})$ such that*

$$\mathcal{T}(F, \mathcal{S}_{\text{in}}, \mathcal{S}_{\text{out}}) := \{(x, y, z) : F(x, y) = z, x \in \mathcal{S}_{\text{in}}, y \in \mathcal{S}_{\text{out}}\} \subseteq \mathcal{C}(F, \mathcal{S}_{\text{in}}, \mathcal{S}_{\text{out}}). \tag{11}$$

When the above assumption holds we shall say that the specification

$$F(x, y) \le 0 \quad \forall x \in \mathcal{S}_{\text{in}}, y = f(x) \tag{12}$$

is convex-relaxable. Assumption 1 means we can find an efficiently computable convex relaxation of the specification which allows us to formulate the verification as a convex optimization problem.

## 3.3 Convex optimization for verification

We can propagate the bounds of our network layer by layer given that the inputs are bounded by $[l_0, u_0]$ thus we can construct the following optimization problem

$$\text{maximize } z \tag{13}$$
$$\text{subject to } (x_0, x_K, z) \in \mathcal{C}(F, \mathcal{S}_{\text{in}}, \mathcal{S}_{\text{out}})$$
$$x_{k+1} \in \text{Relax}(g_k)(W_k x_k + b_k, l_k, u_k), k = 0, \cdots, K - 1$$
$$l_k \le x_k \le u_k, \quad k = 0, \cdots, K$$

over the variables $x, x_1, \cdots, x_k$. Here, $\text{Relax}(g_k)$ represents the relaxed form of the activation function $g_k$. For example, in Bunel et al. (2017); Ehlers (2017) the following convex relaxation of the ReLU$(x) = \max(x, 0)$ function for input $x \in [l, u]$ was proposed:

$$\text{Relax}(\text{ReLU})(x, l, u) = \begin{cases} x & \text{if } l \ge 0 \\ 0 & \text{if } u \le 0 \\ \left\{ \alpha : \alpha \ge x, \alpha \ge 0, \alpha \le \frac{u}{u-l}(x - l) \right\} & \text{otherwise.} \end{cases} \tag{14}$$

---

[1]In all the examples in this paper, we can construct this function in time at most quadratic in $n, m$.

For simplicity, we use $g_k = \text{ReLU}$ and the above relaxation throughout, for other activation functions see Dvijotham et al. (2018b).

**Lemma 3.1.** *Given a neural network $f$, input set $\mathcal{S}_{\text{in}}$, and a convex-relaxable specification $F$, we have that $F(x, y) \leq 0$ for all $x \in \mathcal{S}_{\text{in}}$, $y = f(x)$ if the optimal value of (13) is smaller than $0$ and further (13) it is a convex optimization problem.*

*Proof.* This follows from Assumption 1, for details we refer to Appendix A. $\square$

### 3.4 Modeling specifications in a convex-relaxable manner

**Specifications involving the Softmax.** The specifications on label semantics from (1) can be modeled as linear functions of the softmax layer of the neural network: $P(j|x) = e^{y_j} / \sum_k e^{y_k}$, $y = f(x)$. The semantic distance constraint (1) can then be written as

$$\sum_j P(j|x) d_j \leq \epsilon, \tag{15}$$

where we use $d_j = d(i, j)$ for brevity, and $d_i = 0$. Given initial constraints on the input $x$ (of the form $x \in \mathcal{S}_{\text{in}} = \{x : l_0 \leq x \leq u_0\}$), we use bound propagation techniques to obtain bounds on $\mathcal{S}_{\text{out}} = \{y : l_K \leq y \leq u_K\}$ (as described in Section (Bunel et al., 2017; Dvijotham et al., 2018b)). Thus the specification can be rephrased as

$$F(x, y) = \sum_j \exp(y_j)(d_j - \epsilon) \leq 0 \quad \forall x \in \mathcal{S}_{\text{in}}, y \in \mathcal{S}_{\text{out}}. \tag{16}$$

The set $\mathcal{C}(F, \mathcal{S}_{\text{in}}, \mathcal{S}_{\text{out}})$ can be constructed by noting that the exponential function is bounded above by the linear interpolation between $(l_K, \exp(l_K))$ and $(u_K, \exp(u_K))$. This is given by

$$G(y_i, l_K, u_K) = \frac{\exp(u_{K,i}) - \exp(l_{K,i}))}{u_{K,i} - l_{K,i}} y_i + \frac{u_{K,i} \exp(l_{K,i}) - l_{K,i} \exp(u_{K,i})}{u_{K,i} - l_{K,i}}.$$

We propose the following convex relaxation of the $\exp$ function:

$$\text{Relax}(\exp)(y, l_K, u_K) = \{\alpha : \exp(y) \leq \alpha \leq G(y, l_K, u_K)\}. \tag{17}$$

Given the above relaxation it can be shown (refer to Appendix B), the set below satisfies (11):

$$\mathcal{C}_{\text{smax}}(F, \mathcal{S}_{\text{in}}, \mathcal{S}_{\text{out}}) = \left\{ (x, y, z) : \quad \exists \alpha \text{ s.t.} \quad \begin{array}{l} z = \sum_j \alpha_j (d_j - \epsilon), \\ \alpha \in \text{Relax}(\exp)(y, l_K, u_K) \\ y \in [l_K, u_K] \end{array} \right\} \tag{18}$$

A similar procedure can be extended to the downstream specifications stated in (5).

**Specifications involving quadratic functions.** The energy specification (3) involves a quadratic constraint on both inputs and outputs of the network. Here we show how general quadratic constraints are convex relaxable. Consider an arbitrary quadratic specification on the input and output of the network:

$$F(x, y) = \begin{bmatrix} 1 \\ x \\ y \end{bmatrix}^T Q \begin{bmatrix} 1 \\ x \\ y \end{bmatrix} \leq 0 \qquad \forall x \in \mathcal{S}_{\text{in}}, \ y \in \mathcal{S}_{\text{out}}, \tag{19}$$

where $Q \in \mathbb{R}^{(1+n+m) \times (1+n+m)}$ is a fixed matrix. This can be rewritten as $F(x, y) = \mathbf{Tr}(QX)$ where $X = \alpha \alpha^T$, $\mathbf{Tr}(X) = \sum_i^n X_{ii}$, and $\alpha^T = \begin{bmatrix} 1 & x^T & y^T \end{bmatrix}$. Note that $X_{ij} = \alpha_i \alpha_j$, and we have bounds on $x$ and $y$ which implies bounds on $\alpha$. We denote these bounds as $l_i \leq \alpha_i \leq u_i$ with $l_0 = u_0 = 1$. The set $\{(x, y, z) : F(x, y) = z, x \in \mathcal{S}_{\text{in}}, y \in \mathcal{S}_{\text{out}}\}$ can be rewritten as

$$\left\{ (x, y, z) : z = \mathbf{Tr}(QX), X = \alpha \alpha^T, l \leq \alpha \leq u, \alpha^T = \begin{bmatrix} 1 & x^T & y^T \end{bmatrix} \right\}.$$

Further we note that along the diagonal, we have the functional form $X_{ii} = \alpha_i^2$. This quadratic form is bounded above by the linear interpolation between $(l, l^2)$ and $(u, u^2)$ which is:

$$G_{\text{Quad}}(\alpha, l, u) = (l + u)\alpha - ul.$$

We can relax the constraint $X = \alpha\alpha^T$ as follows (for more details see Appendix C):

$$\text{Relax}\,(\text{Quad})\,(\alpha, l, u) = \left\{ X : \begin{array}{c} X - l\alpha^T - \alpha l^T + ll^T \geq 0 \\ X - u\alpha^T - v\alpha^T + uu^T \geq 0 \\ X - l\alpha^T - \alpha u^T + lu^T \leq 0 \\ \alpha^2 \leq \text{diag}(X) \leq G_{\text{Quad}}(\alpha, l, u) \\ X - \alpha\alpha^T \succeq 0 \end{array} \right\} \quad (20)$$

where the notation $A \succeq B$ is used to indicate that $A - B$ is a *symmetric positive semidefinite* matrix. Given the above relaxation we can show (refer to Appendix C) that the set below satisfies (11):

$$\mathcal{C}_{\text{quad}}\,(F, \mathcal{S}_{\text{in}}, \mathcal{S}_{\text{out}}) = \left\{ (x, y, z) : \exists X, \alpha \text{ s.t} \begin{array}{c} z = Tr(QX) \\ X \in \text{Relax}\,(\text{Quad})\,(\alpha, l, u) \\ \alpha^T = \begin{bmatrix} 1 & x^T & y^T \end{bmatrix} \end{array} \right\}. \quad (21)$$

## 4 EXPERIMENTS

We have proposed a novel set of specifications to be verified as well as new verification algorithms that can verify whether these specifications are satisfied by neural networks. In order to validate our contributions experimentally, we perform two sets of experiments: the first set of experiments tests the ability of the convex relaxation approach we develop here to verify nonlinear specifications and the second set of experiments tests that the specifications we verify indeed provide useful information on the ability of the ML model to solve the task of interest.

**Tightness of verification:** We proposed incomplete verification algorithms that can produce conservative results, i.e., even if the model satisfies the specification our approach may not be able to prove it. In this set of experiments, we quantify this conservatism on each of the specifications from Section 2. In order to do this, we compare the results from the verification algorithm with those from the gradient based falsification algorithm (8) and other custom baselines for each specification. We show that our verification algorithm can produce tight verification results, i.e., the fraction of instances on which the verification algorithm fails to find a proof of a correctness but the falsification algorithm cannot find a counter-example is small. In other words, empirically our strategy is able to verify most instances for which the property to be verified actually holds.

**Value of verification:** In this set of experiments, we demonstrate the effectiveness of using verification as a tool to detect failure modes in neural networks when the specifications studied are not satisfied. For each of the specifications described in Section 2, we compare two models A and B which satisfy our specification to varying degrees (see below and Appendix D for the details). We show that models which violate our convex-relaxed specifications more can exhibit interesting failure modes which would otherwise be hard to detect.

### 4.1 EVALUATION METRICS

Ideally, we would like to verify that the specification is true for all "reasonable" inputs to the model. However, defining this set is difficult in most applications - for example, what is a reasonable input to a model recognizing handwritten digits? Due to the difficulty of defining this set, we settle for the following weaker verification: the specification is true over $\mathcal{S}_{\text{in}}(x^{\text{nom}}, \delta) = \{x : \| x - x^{\text{nom}} \|_\infty \leq \delta\}$, where $x^{\text{nom}}$ is a point in the test set. Since the test set consists of valid samples from the data generating distribution, we assume that inputs close to the test set will also constitute reasonable inputs to the model. We then define the following metrics which are used in our experiments to gauge the tightness and effectiveness our verification algorithm:
**Verification bound:** This is the fraction of test examples $x^{\text{nom}}$ for which the specification is provably satisfied over the set $\mathcal{S}_{\text{in}}(x^{\text{nom}}, \delta)$ using our verification algorithm.
**Adversarial bound:** This is the fraction of test examples $x^{\text{nom}}$ for which the falsification algorithm based on (8) was *not able to find a counter-example in the set* $\mathcal{S}_{\text{in}}(x^{\text{nom}}, \delta)$.
The real quantity of interest is the fraction of test examples for which the specification is satisfied - denoted as $\beta$. We would like the verification bound to be close to $\beta$. However, since $\beta$ is difficult to compute (due to the intractability of exact verification), we need a measurable proxy. To come up

with such a proxy, we note that the verification bound is always lower than the adversarial bound, as the attack algorithm would not be able to find a counter-example for any test example that is verifiable. Formally, the following holds:

$$\text{Verification bound} \leq \beta \leq \text{Adversarial bound}.$$

Thus $|\text{Verification bound} - \beta| \leq \text{Adversarial bound} - \text{Verification bound}$. Hence, we use the difference between the Adversarial bound and the Verification bound to gauge the tightness of our verification bound.

## 4.2 EXPERIMENTAL SETUP

For each of the specifications we trained two networks (referred to as model A and B in the following) that satisfy our specification to varying degrees. In each case model A has a higher verification bound than model B (i.e, it is consistent with the specification over a larger fraction of the test set) – while both networks perform similarly wrt. predictive accuracy on the test set. The experiments in Section 4.3 only use model A while experiments in Section 4.4 uses both model A and model B. In brief (see the Appendix D for additional details):

**MNIST and CIFAR-10:** For these datasets both models was trained to maximize the log likelihood of true label predictions while being robust to adversarial examples via the method described in (Wong & Kolter, 2018). The training for model A places a heavier loss than model B when robustness measures are violated.

**Mujoco:** To test the energy specification, we used the Mujoco physics engine (Todorov et al., 2012) to create a simulation of a simple pendulum with damping friction. We generate simulated trajectories using Mujoco and use these to train a one-step model of the pendulum dynamics. Model B was trained to simply minimize $\ell_1$ error of the predictions while model A was trained with an additional penalty promoting conservation of energy.

## 4.3 TIGHTNESS OF VERIFICATION

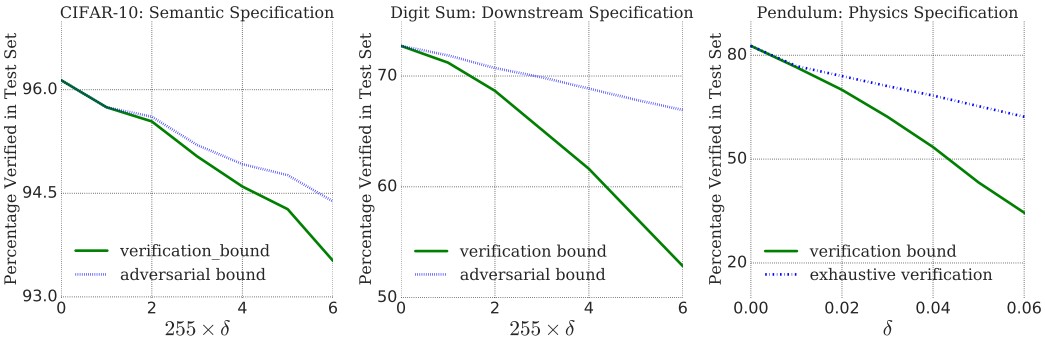

Figure 1: For three specifications, we plot the verification bound and adversarial bound as a function of perturbation size on the input.

**Semantic specifications on CIFAR-10.** We study the semantic distance specification (1) in the context of the CIFAR-10 dataset. We define the distances $d(i, j)$ between labels as their distance according to Wordnet (Miller, 1995) (the full distance matrix used is shown in Appendix F). We require that the expected semantic distance from the true label under an adversarial perturbation of the input of radius $\delta$ is smaller than $\epsilon = .23$. The threshold 0.23 was chosen so that subcategories of man-made transport are semantically similar and subcategories of animals are semantically similar. The results are shown in Fig. 1 (left). For all $\delta \leq 6/255$, the gap between the adversarial and verification bound is smaller than .9%.

**Errors in a system predicting sums of digits.** We study a specification of the type (5) with $N = 2$ and $\epsilon = 1$. The results are shown in Figure 1 (middle). For $\delta \leq 2/255$, the gap between the adversarial and verification bound is less than 1.5%. At $\delta = 6/255$, the gap increases to about 13%. We suspect this is because of the product of softmax probabilities, that degrades the tightness of the

relaxation. Tighter relaxations that deal with products of probabilities are in interesting direction for future work - signomial relaxations (Chandrasekaran & Shah, 2016) may be useful in this context.

**Conservation of energy in a simple pendulum.** We study energy conservation specification in a damped simple pendulum, as described in (3) and learn a dynamics model $x_{t+1} = f(x_t)$. Since the state $x_t$ is only three dimensional (horizontal position, vertical position and angular velocity), it is possible to do exhaustive verification (by discretizing the state space with a very fine grid) in this example, allowing us to create a "golden baseline". Note that this procedure is computationally expensive and the brute force search is over 239000 points. The results are shown in Fig. 1 (right). At a perturbation radius up until 0.02, the exhaustive verification can prove that the specification is not violated for 73.98% of the test data points while using our convex relaxation approach achieved a lower bound of 70.00%. As we increase the size of perturbation up to 0.06, the gap between the two increases to 22.05%. However, the convex relaxation runs an order of magnitude faster given the same computational resources - this gap in computation time would increase even further as the state dimension grows.

### 4.4 VALUE OF VERIFICATION

Consider the following scenario: We are given a pair of ML models for a task - model A and model B. We do not necessarily know how they have been trained but would like to evaluate them before deploying them. Standard evaluation in terms of predictive performance on a hold-out test set cannot distinguish between the models. We consider using verification of a specification relevant to the task as an additional way to evaluate these models. If model A is more verifiable (the verification bound is higher) than the other, we hope to show that model A is better at solving the task than model B. The following experiments demonstrate this for each of the specifications studied in this paper, demonstrating the value of verification as a diagnostic tool for tasks with complex specifications.

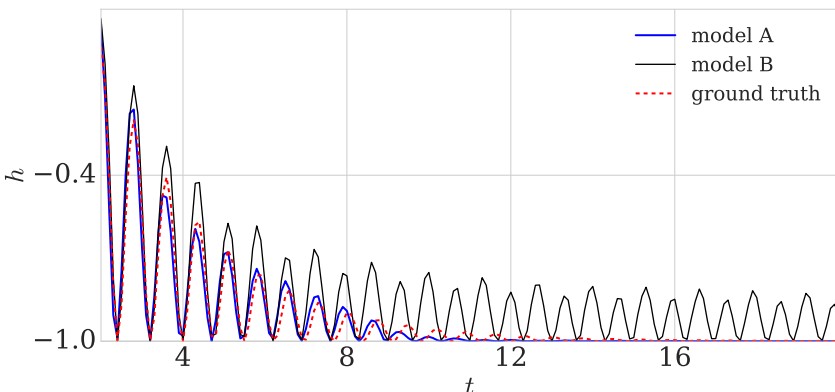

Figure 2: The red dashed line displays the pendulum's ground truth trajectory (height vs time). The black line shows a rollout from model B, the blue line shows a rollout from model A. The more verifiable model (model A) clearly tracks the ground truth trajectory significantly more closely.

**Energy specifications and cascading errors in model dynamics:** Models A and B were both trained to learn dynamics of a simple pendulum (details in Appendix D). At $\delta = .06$ [2], the verification bound (for the energy specification (3)) for model A is 64.16% and for model B is 34.53%. When a model of a dynamical systems makes predictions that violate physical laws, its predictions can have significant errors over long horizons. To evaluate long-term predictive accuracy of the two models, we rollout of a long trajectory (200 steps) under models A, B and compare these to the ground truth (a simulation using the actual physics engine Mujoco). The results are shown in Fig. 2. For model A the rolled out trajectory matches the ground truth much more closely. Further, the trajectory from model A eventually stabilizes at the vertical position (a stable equilibrium for the pendulum) while that from model B keeps oscillating.

---

[2]which is equivalent to perturbing the angle of the pendulum by $\theta = 0.04$ radians and the angular velocity by $0.6 m/s$

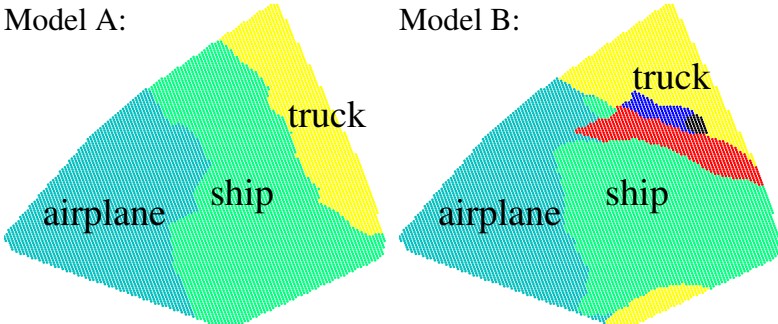

Figure 3: The projection of the decision boundaries onto a two dimensional surface formed by interpolating between three images belonging to the same semantic category (vehicles) - aeroplane (cyan), ship (green) and truck (yellow).The red/blue/**black** regions represent bird/cat/**frog** respectively).

**Expected semantic distance.** Here, we verify models A and B with respect to the semantic specification (15) and a verification bound of 93.5% for model A and 84.3% for model B. Thus, we expect that model B will be more susceptible to produce labels from semantically dissimilar categories than model A under small perturbations of the input. To show this, we visualize the decision boundaries of the two models projected onto a 2-D surface created by interpolating between three images which are in semantically similar categories (man-made transportation). Fig. 3 shows the decision boundaries. Indeed, we can observe that the model with a higher verification bound displays higher semantic consistency in the decision boundaries.

## 5    CONCLUSIONS

We have developed verification algorithms for a new class of *convex-relaxable specifications*, that can model many specifications of interest in prediction tasks (energy conservation, semantic consistency, downstream task errors). We have shown experimentally that our verification algorithms can verify these specifications with high precision while still being tractable (requiring solution of a convex optimization problem linear in the network size). We have shown that verifying these specifications can indeed provide valuable diagnostic information regarding the ultimate behavior of models in downstream prediction tasks. While further work is needed to scale these algorithms to real applications, we have made significant initial steps in this work. Further, inspired by (Wong & Kolter, 2018; Wong et al., 2018; Raghunathan et al., 2018), we believe that folding these verification algorithms can help significantly in training models that are consistent with various specifications of interest. We hope that these advances will enable the development of general purpose specification-driven AI, where we have general purpose systems that can take specifications stated in a high level language along with training data and produce ML models that achieve high predictive accuracy while also being consistent with application-driven specifications. We believe that such an advance will significantly aid and accelerate deployment of AI in applications with safety and security constraints.

## ACKNOWLEDGEMENTS

We thank Jost Tobias Springenberg and Jan Leike for careful proof-reading of this paper.

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

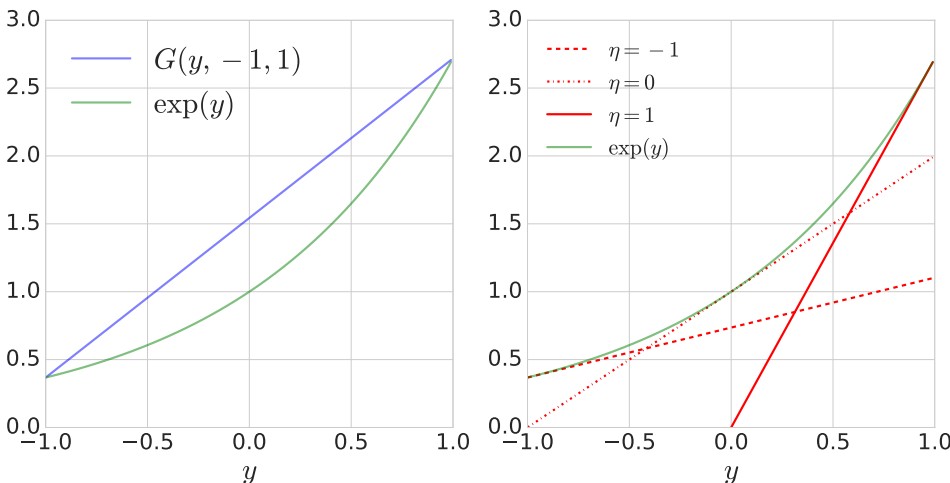

Figure 4: This plots the convex constraint $G(y, 1, -1)$ (left) and the tangential constraints for an exponential function when $y \in [l, u]$ where $l = -1$ and $u = 1$.

## A  PROOF FOR LEMMA 3.1

This can be done by proof of contradiction. Suppose that the optimal value for $\mathcal{C}\left(F, \mathcal{S}_{\text{in}}, \mathcal{S}_{\text{out}}\right)$ is less than zero, but there exists a point in the set

$$(x_F, y_F, z_F) \in \mathcal{T}\left(F, \mathcal{S}_{\text{in}}, \mathcal{S}_{\text{out}}\right)$$

such that $F(x_F, y_F) > 0$, then by definition $(x_F, y_F, z_F) \in \mathcal{C}\left(F, \mathcal{S}_{\text{in}}, \mathcal{S}_{\text{out}}\right)$ which gives a contradiction. Further we note that the objective in (13) is linear and the constraints imposed form a convex set thus it is a convex optimization problem.

## B  SOFTMAX RELAXATION

We outline in detail the convex relaxations we used for the softmax function to obtain the set $C_{\text{smax}}(F, \mathcal{S}_{\text{in}}, \mathcal{S}_{\text{out}})$. Here we note that the denominator of a softmax cannot be zero as it is an exponential function over the real line, thus the following is true:

$$F(x, y) = \sum_i d_i \frac{\exp(y_i)}{\sum_j \exp(y_j)} = z \Leftrightarrow \sum_i (d_i - z) \exp(y_i) = 0.$$

$$\Rightarrow \mathcal{T}(F, \mathcal{S}_{\text{in}}, \mathcal{S}_{\text{out}}) = \left\{ (x, y, z) : \sum_i (d_i - z) \exp(y_i) = 0, x \in \mathcal{S}_{\text{in}}, l_K \leq y \leq u_K \right\}.$$

For the relaxation of the exponential function on auxiliary variable $\alpha$, we note that the convex constraint is the linear interpolation between $(l_{K,i}, \exp(l_{K,i}))$ and $(u_{K,i}, \exp(u_{K,i}))$ shown in Fig. 4. This linear interpolation is given by the following:

$$G(y_i, l_K, u_K) = \frac{\exp(u_{K,i}) - \exp(l_{K,i})}{u_{K,i} - l_{K,i}} y_i + \frac{u_{K,i} \exp(l_{K,i}) - l_{K,i} \exp(u_{K,i})}{u_{K,i} - l_{K,i}}.$$

The bounds $\exp(y_i) \leq \alpha \leq G_U(y_i)$ defines a convex set on $\alpha$, further we note that

$$\exp(y_i) \leq \exp(y_i) \leq G_U(y_i).$$

Thus, the following holds

$$\mathcal{T}(F, \mathcal{S}_{\text{in}}, \mathcal{S}_{\text{out}}) \subseteq \mathcal{C}_{\text{smax}}(F, \mathcal{S}_{\text{in}}, \mathcal{S}_{\text{out}}) \quad \square$$

For implementation of the lower bound $\exp(y_i) \leq \alpha_i$, we transform this bound into a set of linear tangential constraints (shown in Fig. 4), such that it can be solved with an LP. The tangent at point $\eta \in [l, u]$ is given by

$$G_L(y_i, \eta) = \exp(\eta) y_i + (\exp(\eta) - \eta \exp(\eta))$$

This allows us to relax the exponential function by the following set of linear constraints:

$$G_L(y_i, \eta) \leq \alpha \leq G_U(y_i) \quad \forall \eta \in [l_i, u_i].$$

## C  QUADRATIC RELAXATION

Here we outline in detail the convex relaxation of the quadratic function to obtain the set $C_{\text{quad}}(F, \mathcal{S}_{\text{in}}, \mathcal{S}_{\text{out}})$. The quadratic function is of the form

$$F(x, y) = \begin{pmatrix} 1 \\ x \\ y \end{pmatrix}^T Q \begin{pmatrix} 1 \\ x \\ y \end{pmatrix}.$$

For simplicity of notation we will use $\alpha = (1, x, y)$, we denote its ith element as $x_i$. For this particular case, the specification can be written as $F(x, y) = \sum_{ij} Q_{ij}\alpha_i\alpha_j$. Now we derive our relaxation for the quadratic function, $\text{Relax}(\text{Quad})(\alpha, l, u)$. With the constraints we place on $\alpha$: $l_i \leq \alpha_i \leq u_i$, we can use the McCormick envelopes

$$
\begin{aligned}
(\alpha_i - l_j)(\alpha_i - u_i) &\leq 0 \\
(\alpha_j - u_j)(\alpha_i - l_i) &\leq 0 \\
(\alpha_j - u_j)(\alpha_i - u_i) &\geq 0 \\
(\alpha_j - l_j)(\alpha_i - l_i) &\geq 0
\end{aligned}
\tag{22}
$$

to obtain constraints on the quadratic function. Given (22) we can relax $X = \alpha\alpha^T$ with the following set of constraints:

$$
\begin{aligned}
X_{ij} - l_j\alpha_i - u_i\alpha_j + l_j u_i &\leq 0 \\
X_{ij} - l_i\alpha_j - u_j\alpha_i + l_i u_j &\leq 0 \\
X_{ij} - u_i\alpha_i - u_j\alpha_j + u_i u_j &\geq 0 \\
X_{ij} - l_i\alpha_i - l_j\alpha_j + l_i l_j &\geq 0.
\end{aligned}
$$

We can enforce extra constraints down the diagonal of the matrix $X$, as the diagonal is of the form $X_{ii} = \alpha_i^2$. Since this is a convex function on the auxiliary variable $\alpha_i$, we can again use the convex constraint which is the linear interpolation between points $(l, l^2)$ and $(u, u^2)$, given by:

$$G_{\text{Quad}}(\alpha, l, u) = (l + u)\alpha - ul$$

Thus the following denotes a valid convex set on the diagonal of $X$:

$$\alpha^2 \leq \text{diag}(X) \leq G_{\text{Quad}}(\alpha, l, u)$$

Further we note that $X_{ij}$ is a symmetric semi-positive definite matrix, we can impose the following constraint (see Luo et al. (2010)):

$$X - \alpha\alpha^T \succeq 0.$$

Explicitly we enforce the constraint:

$$\begin{pmatrix} 1 & \alpha^T \\ \alpha & X \end{pmatrix} \succeq 0 \tag{23}$$

A thing to note is that constraints

$$
\begin{aligned}
X_{ij} - l_j\alpha_i - u_i\alpha_j + l_j u_i &\leq 0 \\
X_{ij} - l_i\alpha_j - u_j\alpha_i + l_i u_j &\leq 0,
\end{aligned}
$$

becomes the same constraint when $X - \alpha\alpha^T \succeq 0$ is enforced, since $X$ is a symmetric matrix thus one was dropped in (20). We note that $\alpha\alpha^T \in \text{Relax}(\text{Quad})(X, l, u)$, therefore the following holds:

$$\mathcal{T}(F, \mathcal{S}_{\text{in}}, \mathcal{S}_{\text{out}}) \subseteq \mathcal{C}_{\text{quad}}(F, \mathcal{S}_{\text{in}}, \mathcal{S}_{\text{out}}) \quad \square$$

## D  TRAINING DETAILS

This paper is primarily focused on new techniques for verification rather than model training. However, generally speaking, training neural networks using standard methods does not produce verifiable models. In fact, it has been observed that if one trains networks using standard cross entropy loss or even using adversarial training, networks that seem robust empirically are not easily verified

due to the incomplete nature of verification algorithms (Wong & Kolter, 2018). In Wong & Kolter (2018); Wong et al. (2018); Gowal et al. (2018), the importance of training networks with a special loss function that promotes verifiability was shown when attempting to obtain verifiably robust networks (against $\ell_\infty$ adversarial perturbations). Similar observations have been made in (Gehr et al., 2018; Raghunathan et al., 2018). The specifications we study in this paper (1,3, 5) build upon the standard adversarial robustness specification. Hence, we use the training method from Wong et al. (2018) (which has achieved state of the art verifiable robustness against $\ell_\infty$ adversarial perturbations on MNIST and CIFAR-10) to train networks for our CIFAR-10 and MNIST experiments.

**Cifar 10: Model A**  We use a network that is verifiably robust to adversarial pertubations of size $8/255$ (where 255 is the range of pixel values) on $24.61\%$ of the test examples, with respect to the standard specification that the output of the network should remain invariant to adversarial perturbations of the input. The network consists of 4 convolutional layers and 3 linear layers in total 860000 paramters.

**Cifar 10: Model B**  We use a network that is verifiably robust to adversarial pertubations of size $2/255$ (where 255 is the range of pixel values) on $39.25\%$ of the test examples, with respect to the standard specification that the output of the network should remain invariant to adversarial perturbations of the input. The architecture used is the same as Model A above.

**Mujoco**  We trained a network using data collected from the Mujoco simulator (Todorov et al., 2012), in particular, we used the pendulum swingup environment in the DM control suite (Tassa et al., 2018). The pendulum is of length 0.5m and hangs 0.6m above ground. When the perturbation radius is 0.01, since the pendulum is 0.5m in length, the equivalent perturbation in space is about 0.005 m in the x and y direction. The perturbation of the angular velocity is $\omega \pm 0.1$ radians per second.

The pendulum environment was used to generate 90000 (state, next state) pairs, 27000 was set aside as test set. For training the timestep between the state and next state pairs is chosen to be 0.1 seconds (although the simulation is done at a higher time resolution to avoid numerical integration errors).

The pendulum models consists of two linear layers in total 120 parameters and takes $(\cos(\theta), \sin(\theta), \omega/10)$ as input. Here $\theta$ is the angle of the pendulum and $\omega$ is the angular velocity. The data is generated such that the initial angular velocity lies between (-10, 10), by scaling this with a factor of 10 ($\omega/10$) we make sure $(\cos(\theta), \sin(\theta), \omega/10)$ lies in a box where each side is bounded by [-1, 1].

**Pendulum: Model A**  We train with an $\ell_1$ loss and energy loss on the next state prediction, the exact loss we impose this model is (we denote $(w_T, h_T, s\omega_T)$ as ground truth state):

$$l(f) = \underbrace{\| f(w, h, s\omega) - (w_T, h_T, s\omega_T) \|}_{\ell_1 \text{ loss}} + \underbrace{\lfloor E(f(w, h, s\omega)) - E((w_T, h_T, s\omega_T)) \rfloor}_{\text{energy difference loss}} + \quad (24)$$

$$\underbrace{\text{ReLU}(E(f(w, h, s\omega)) - E(w, h, s\omega))}_{\text{increase in energy loss}}, \quad (25)$$

where $s = 0.1$ is a scaling parameter we use on angular velocity. $E$ is given in (2). The loss we calculate on the test set is of the $\ell_1$ loss only and we obtain 0.072.

**Pendulum: Model B**  We train with an $\ell_1$ loss on the next state prediction

$$l(f) = \| f(w, h, s\omega) - (w_T, h_T, s\omega_T) \|,$$

The $\ell_1$ loss on the test set is 0.054.

**Digit Sum: Model A**  We use a network, consisting of two linear layers in total 15880 parameters. This network is verifiably robust to adversarial pertubations of size $25/255$ (where 255 is the range of pixel values) on $68.16\%$ of the test examples with respect to the standard specification. The standard specification being that the output of the network should remain invariant to adversarial perturbations of the input.

**Falsification:** The falsification method (8) is susceptible to local optima. In order to make the falsification method stronger, we use 20 random restarts to improve the chances of finding better local optima.

## E    SCALING AND IMPLEMENTATION:

The scaling of our algorithm is dependent on the relaxations used. If the relaxations are a set of linear constraints, the scaling is linear with respect to the input and output dimensions of the network, if they are quadratic constraints they will scale quadratically.

For the Semantic Specification and Downstream Specification, the relaxations consisted of only linear constraints thus can be solved with a linear program (LP). For these tasks, we used GLOP as the LP solver. This solver takes on approximately 3-10 seconds per data point on our desktop machine (with 1 GPU and 8G of memory) for the largest network size we handled which consists of 4 convolutional layers and 3 linear layers and in total 860000 parameters. For the conservation of energy, we used SDP constraints - this relaxation scales quadratically with respect to the input and output dimension of the network. To solve for these set of constraints we used CVXPY, which is much slower than GLOP, and tested on a smaller network which has two linear layers with 120 parameters.

## F    DISTANCES BETWEEN CIFAR-10 LABELS IN WORDNET

The synsets chosen from WordNet are: airplane.n.01, car.n.01, bird.n.01, cat.n.01, deer.n.01, dog.n.01, frog.n.01, horse.n.01, ship.n.01, truck.n.01. The path similarity, here we denote as $x$, in WordNet is between [0, 1/3], to make this into a distance measure between labels which are not similar we chose $d = 1/3 - x$.

|            | automobile | bird | cat  | deer | dog  | frog | horse | ship | truck |
|------------|------------|------|------|------|------|------|-------|------|-------|
| airplane   | 0.22       | 0.27 | 0.28 | 0.28 | 0.26 | 0.27 | 0.28  | 0.17 | 0.22  |
| automobile | –          | 0.26 | 0.28 | 0.28 | 0.26 | 0.27 | 0.28  | 0.21 | 0.00  |
| bird       |            | –    | 0.19 | 0.21 | 0.17 | 0.08 | 0.21  | 0.26 | 0.26  |
| cat        |            |      | –    | 0.21 | 0.13 | 0.21 | 0.21  | 0.28 | 0.28  |
| deer       |            |      |      | –    | 0.21 | 0.22 | 0.19  | 0.28 | 0.28  |
| dog        |            |      |      |      | –    | 0.19 | 0.21  | 0.26 | 0.26  |
| frog       |            |      |      |      |      | –    | 0.22  | 0.27 | 0.27  |
| horse      |            |      |      |      |      |      | –     | 0.28 | 0.28  |
| ship       |            |      |      |      |      |      |       | –    | 0.21  |

Table 1: This is the distance matrix $d(y, y')$ used for CIFAR-10.

## G CIFAR-10: SEMANTIC DECISION BOUNDARIES

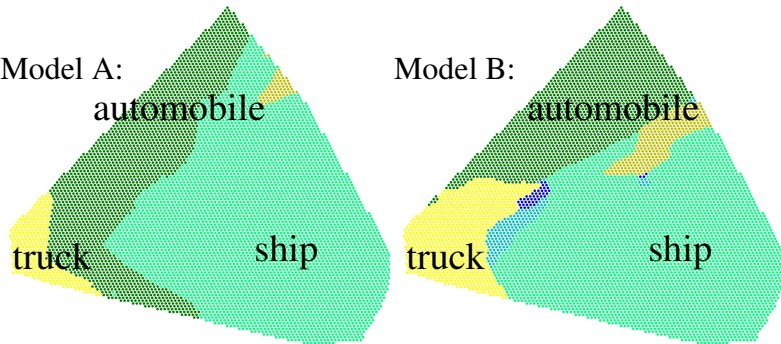

Figure 5: We plot the projection of the decision boundaries onto a two dimensional surface formed by interpolating between three images belonging to the same semantic category (vehicles) - truck (yellow), ship (green) and automobile (dark green). cyan/blue regions represent airplane/cat.

## H COMPARISON OF TIGHTNESS

For verification purposes, it is essential to find a convex set which is sufficiently tight such that the upper bound found on $F(x, y)$ is a close representative of $\max_{x \in S_{\text{in}}} F(x, f(x))$. We use the difference between the verification bound and adversarial bound as a measure of how tight the verification algorithm is. More concretely, the difference between the adversarial bound and verification bound should decrease as the verification scheme becomes tighter. Here, we compare the tightness of two different convex-relaxations for the physics specification on the pendulum model (Model B).

The SDP convex relaxation (the one we have shown in the main paper) uses the following relaxation of the quadratic function:

$$\text{Relax}\,(\text{Quad})\,(\alpha, l, u) = \left\{ X : \begin{array}{l} X - l\alpha^T - \alpha l^T + ll^T \geq 0 \\ X - u\alpha^T - v\alpha^T + uu^T \geq 0 \\ X - l\alpha^T - \alpha u^T + lu^T \leq 0 \\ \alpha^2 \leq \text{diag}(X) \leq G_{\text{Quad}}(\alpha, l, u) \\ X - \alpha\alpha^T \succeq 0 \end{array} \right\}, \tag{26}$$

To enforce the constraint $X - \alpha\alpha^T \succeq 0$ means that the complexity of verification increases quadratically with respect to the dimension of $\alpha$. Thus, we can trade complexity for a looser convex-relaxation of the quadratic function given by following:

$$\widehat{\text{Relax}}\,(\text{Quad})\,(\alpha, l, u) = \left\{ X : \begin{array}{l} X - l\alpha^T - \alpha l^T + ll^T \geq 0 \\ X - u\alpha^T - v\alpha^T + uu^T \geq 0 \\ X - l\alpha^T - \alpha u^T + lu^T \leq 0 \\ X - u\alpha^T - \alpha l^T + lu^T \leq 0 \\ \alpha^2 \leq \text{diag}(X) \leq G_{\text{Quad}}(\alpha, l, u) \end{array} \right\}. \tag{27}$$

Comparing both relaxations can give us a gauge of how much enforcing the SDP constraint $X - \alpha\alpha^T \succeq 0$ gives us in practise. The results are shown in Fig. 6. What we see is that enforcing the SDP constraints at a small perturbation radius (i.e. $\delta \leq 0.02$), gives us a verification bound which allows us to verify only $0.9\%$ more test points. As the perturbation radius increases SDP constraint allowed us to verify $4.8\%$ more test data points than without. In this case, these results suggest that the perturbation radius $\delta$, which is less or equal to 0.06, is sufficiently small such that enforcing higher order constraints makes for only slightly tighter convex-relaxations. In general it is often the case that we can enforce the cheapest (computationally) relaxations if our input set $S_{\text{in}}$ is sufficiently small.

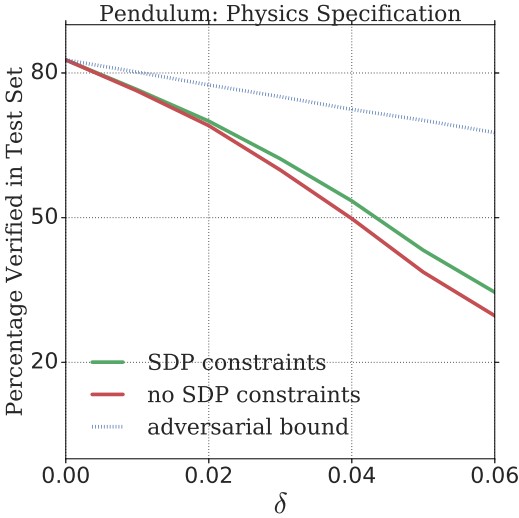

Figure 6: We compared two different convex-relaxations of the physics specification. Specifically, one which is with SDP constraints shown in Eq. (23) and one without the SDP constraint.

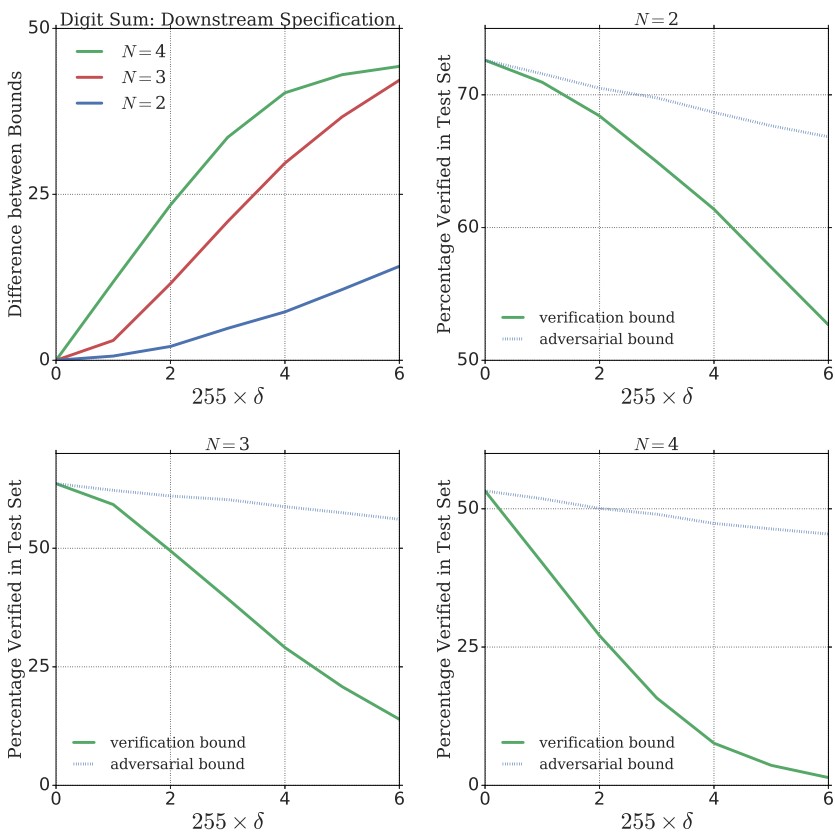

Figure 7: Top left plot shows the difference between verification bound and adversarial bound with respect to $N = 2, 3$ and 4, where $N$ is the number of digits summed (see. (4)). All other plots shows the verification bound and the adversarial bound for $N = 2, 3, 4$ digits summed.

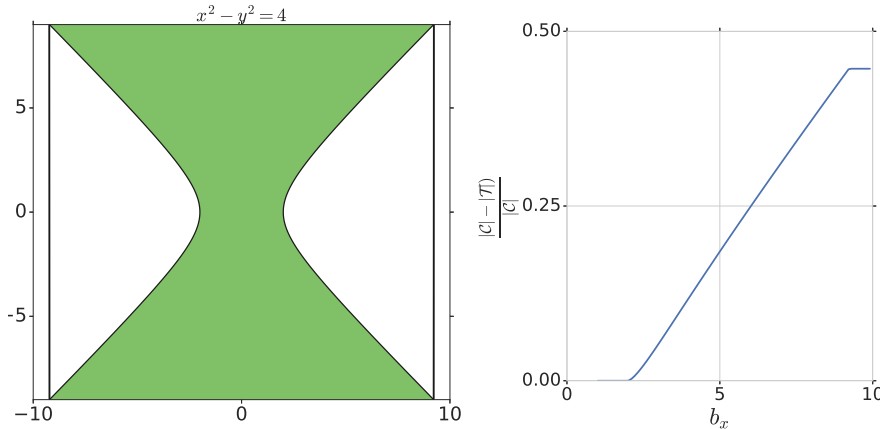

Figure 8: On the left is a plot showing the true feasible region ($\mathcal{T}$) for $x^2 - y^2 \le 4$ (green) where $b_x = 10$ and $b_y = 9$, $S_{\text{in}} = \{x : -b_x \le x \le b_x\}$ and $S_{\text{out}} = \{y : -b_y \le y \le b_y\}$ with its corresponding convex relaxed region $\mathcal{C}$ (black lined box). The plot on the right shows $\frac{|\mathcal{C}| - |\mathcal{T}|}{|\mathcal{C}|}$, where $|\mathcal{C}|$ represents the volume of the set, as $b_x$ increases and $b_y = 9$.

## H.1 DOWNSTREAM SPECIFICATION: TIGHTNESS AS WE INCREASE DIGITS TO SUM

For the digit sum problem, we also investigated into the tightness of the verification bound with respect to the adversarial bound as we increase the number of digits, $N$, which we sum over. We used the same network (Model A) for all results. The results are shown in Fig. 7.

The nominal percentage (namely when the input set $S_{\text{in}}$ consists only of the nominal test point) which satisfies the specification decreases from to 72.6% to 53.3% as $N$ increases from 2 to 4. At the same time the difference (we denote as $\Delta$) between the verification bound and adversarial bound increases as $N$ increases (shown in top left plot). We note that at small perturbation radius where $\delta = 2/255$, $\Delta = 4.2\%$, $14.1\%$, $26.1\%$ for $N = 2, 3, 4$ respectively. The increase in $\Delta$ is linear with respect to increase in $N$ when $\delta$ is sufficiently small. At a larger perturbation radius, as the percentage of test data points verifiable tends towards zero, the rate of increase in $\Delta$ as $N$ increases slows down. Concretely, at $\delta = 6/255$, $\Delta = 19.9\%$, $49.7\%$, $51.8\%$ for $N = 2, 3, 4$. Here, we note that the increase in $\Delta$ going from $N = 3$ to 4 is only 2.1%.

## H.2 TIGHTNESS WITH A TOY EXAMPLE

To give intuition regarding the tightness of a convex-relaxed set, we use a toy example to illustrate. Here we consider the true set, $\mathcal{T}(F, S_{\text{in}}, S_{\text{out}})$ where $S_{\text{in}} = \{x : -b_x \le x \le b_x\}$, $S_{\text{out}} = \{y : -b_y \le y \le b_y\}$ and $F(x, y) = x^2 - y^2 \le 4$ (throughout this example we keep $b_y = 9$). We can find its corresponding convex-relaxed set using $\text{Relax}(\text{Quad})$ which results in the following:

$$\mathcal{C}(F, S_{\text{in}}, S_{\text{out}}) = \left\{ (x, y, z) : \exists X, Y, \alpha_x, \alpha_y \text{ s. t } \begin{array}{l} z = X - Y \\ X \in \text{Relax}(\text{Quad})(\alpha_x, -b_x, b_x) \\ Y \in \text{Relax}(\text{Quad})(\alpha_y, -b_y, b_y) \\ \alpha_x = [1, x], \alpha_y = [1, y] \end{array} \right\}.$$

The sets are depicted in Fig. 8, where we see that $\mathcal{C}$ is simply a box around the true set $\mathcal{T}$ (green). With this toy example we can also ask the question what the difference in volume between the two sets is, as this can give us an idea of the tightness of the convex relaxation. The difference is shown on the right plot of Fig. 8. We can see that as we increase $b_x$, while $b_y = 9$, the fractional difference in volume increases linearly up to when $b_x$ is the same size as $b_y = 9$.

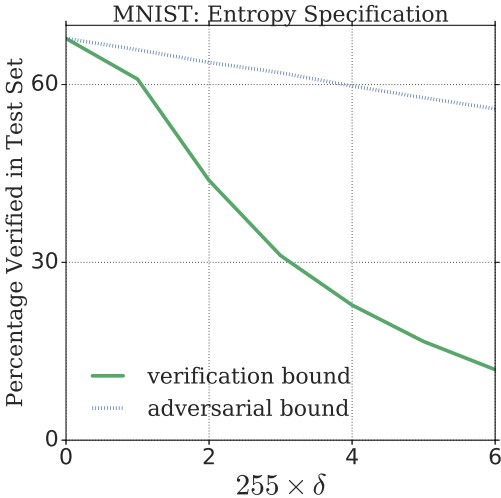

Figure 9: This plot shows the entropy specification, (28), satisfaction for an MNIST classifier where $E = 0.1$.

## I  ENTROPY SPECIFICATION

Another specification which we considered was the entropy specification. Lets consider the scenario, where we would never want our network to be 100% confident about a certain image. That is to say we would like this network to always maintain a level of entropy $E$. Thus for this our specification is the following:

$$F(x, y) = E + \sum_i \frac{\exp(y_i)}{\sum_j \exp(y_j)} \log\left(\frac{\exp(y_i)}{\sum_j \exp(y_j)}\right) \leq 0 \qquad (28)$$

We note that this specification can be rewritten as the following:

$$F(x, y) = \left(\sum_i E \exp(y_i) + y_i \exp(y_i)\right) - \log\left(\sum_j \exp(y_j))\right) \leq 0.$$

Here the relaxations we use is constructed upon $\mathrm{Relax(exp)}$ and $\mathrm{Relax(Quad)}$. Concretely, our relaxation is given by:

$$\mathcal{C} = \left\{ (x, y, z) : \exists\, \alpha_1, X, \alpha_2, Q \text{ s.t. } \begin{array}{l} z = E \sum_i y_i + \mathrm{Trace}(Q_{[:n, n:2n]}) - \log\left(\sum_i \exp(y_i)\right) \\ Q \in \mathrm{Relax(Quad)}(\alpha_2, \tilde{l}_K, \tilde{u}_K) \\ \tilde{l}_K, \tilde{u}_K = [l_K, \exp(l_K)], [u_K, \exp(u_K)] \\ \alpha_2 = [y, X] \\ X \in \mathrm{Relax(exp)}(\alpha_1, l_K, u_K) \\ \alpha_1 = y \end{array} \right\}$$

here $n$ is the number of labels or equivalently the dimensions of the output $y$. We show the results in Fig. 9. The model we have used is a MNIST classifier with two linear layers and 15880 parameters. At a perturbation radius of $\delta = 2/255$, the difference between the verification and adversarial bound is 19.9%, as we increase this to 6 pixel perturbations, the difference between the two bounds becomes close to 44%.

