# OpenReview forum: "Verification of Non-Linear Specifications for Neural Networks"
_ICLR.cc/2019/Conference_

### Official Review · AnonReviewer1 · 2018-11-02
**Some new ideas to generalize verifications for adversarial robustness but limited investigation and experimental results.**

**Rating:** 7
**Confidence:** 3

**Review:**

- Summary: This paper proposes verification algorithms for a class of convex-relaxable specifications to evaluate the robustness of the network under adversarial examples. Experimental results are shown for semantic specifications for CIFAR, errors in predicting sum of two digits and conservation of energy in a simple pendulum.

- Clarity and correctness: It is a well-written and well-organized paper. Notations and expressions are clear. The math seems to be correct.

- Significance: The paper claims to have introduced a class of convex-relaxable specifications which constitute specifications that can be verified using a convex relaxation. However, as described later in the paper, it is limited to feed-forward neural networks with ReLU and softmax activation functions and quadratic parts (it would be better to tone down the claims in the abstract and introduction parts.)

- Novelty: The idea of accounting for label semantics and quadratic expressions when training a robust neural network is important and very practical. This paper introduces some nice ideas to generalize linear verification functions to a larger class of convex-relaxable functions, however, it seems to be more limited in practice than it claims and falls short in presenting justifying experimental results.

** More detailed comments:

** The idea of generalizing verifications to a convex-relaxable set is interesting, however, applying it in general is not very clear -- as the authors worked on a case by case basis in section 3.1.

** One of my main concerns is regarding the relaxation step. There is no discussion on the effects of the tightness of the relaxation on the actual results of the models; when in reality, there is an infinite pool of candidates for 'convexifying' the verification functions. It would be nice to see that analysis as well as a discussion on how much are we willing to lose w.r.t. to the tightness of the bounds -- especially when there is a trade-off between better approximation to the verification function and tightness of the bound.

** I barely found the experimental results satisfying. To find "reasonable" inputs to the model, authors considered perturbing points in the test set. However, I am not sure if this is a reasonable assumption when there would be no access to test data points when training a neural network with robustness to adversarial examples. And if bounding them is a very hard task, I am wondering if that is a reasonable assumption to begin with.

** It is hard to have a sense of how good the results are in Figure 1 due to lack of benchmark results (I could not find them in the Appendix either.)

** The experimental results in section 4.4 are very limited. I suggest that the authors consider running more experiments on more data sets and re-running them with more settings (N=2 for digit sums looks very limited, and if increasing N has some effects, it would be nice to see them or discuss those effects.)

** Page 2, "if they do a find a proof" should be --> "if they do find a proof"
** Page 5, "(as described in Section (Bunel et al., 2017; Dvijotham et al., 2018)", "Section" should be omitted.

******************************************************
After reading authors' responses, I decided to change the score to accept. It got clear to me that this paper covers broader models than I originally understood from the paper. Changing the expression to general forms was a useful adjustment in understanding of its framework. Comparing to other relaxation technique was also an interesting argument (added by the authors in section H in the appendix). Adding the experimental results for N=3 and 4 are reassuring.
One quick note: I think there should be less referring to papers on arxiv. I understand that this is a rapidly changing area, but it should not become the trend or the norm to refer to unpublished/unverified papers to justify an argument.

---

> ### Author Response · Authors · 2018-11-16
> **We thank the reviewer for the detailed feedback**
>
> We thank the reviewer for the detailed feedback and criticism. We made adjustments to the paper to address all your concerns and detail the changes below. We hope the changes clarify the concerns regarding the generality of our algorithm and the requested additional experiments.
>
> Comment 1: [The method is limited to feed-forward neural networks with ReLU and softmax activation functions and quadratic parts (it would be better to tone down the claims in the abstract and introduction parts.)]
>
> Answer 1: We want to clarify that although we have demonstrated most of the results on ReLU feedforward neural networks, it is not limited to such networks. The feedforward nature is indeed required but the ReLU activation function can be replaced with arbitrary activation functions, for example tanh or sigmoid activations (please see https://arxiv.org/abs/1803.06567 for more details). We initially used the ReLU example for clarity of presentation, as a result, maybe the generality of our result is not clear. To address this we have updated Sections 3.1, 3.3 and 3.4. Specifically, we changed the equation: X_{k+1} = ReLU(W_k x_k + b_k) to X_{k+1} = g_k(W_k x_k + b_k). The only change required going from the ReLU equation to the more general equation is the way the bounds ([l_k, u_k]) are propagated through the network and the relaxations applied on the activation functions. For a more general overview of the bound propagation techniques and relaxation of arbitrary activation functions we refer to the following papers https://arxiv.org/abs/1803.06567, https://arxiv.org/pdf/1610.06940.pdf, https://arxiv.org/abs/1805.12514 .
>
> We would also like to clarify that this paper provides a framework for general nonlinear specifications that are convex-relaxable. Although we presented softmax and quadratic specifications this algorithm is not limited to these two cases. To demonstrate this further, we have added Appendix I where we find a convex-relaxation to the entropy of a softmax distribution from a classifier network and use it to verify that a given network is never overly confident. In other words; we would like to verify that a threshold on the entropy of the class probabilities is never violated. The specification at hand is the following:
> F(x, y) = E + \sum_i exp(y_i)/(\sum_j exp(y_j)) log(exp(y_i)/(\sum_j exp(y_j))) <=0
> which is a non-convex function of the network outputs.
>
> Comment 2: Novelty: The idea of accounting for label semantics and quadratic expressions when training a robust neural network is important and very practical. This paper introduces some nice ideas to generalize linear verification functions [...]  it seems to be more limited in practice than it claims and falls short in presenting justifying experimental results.
>
> Answer 2: We emphasize that our method is not limited to quadratic expressions and label semantics and refer to Answer 1, above, for comments regarding the generality. Regarding your concerns wrt the novelty of the approach: as far as we are aware there is no prior paper considering the problem of verifying nonlinear specifications for neural networks. Regarding the presentation of results: We refer to Answer 6, below, for a detailed justification of our experimental procedure. Additionally we want to highlight that our verification tool was a useful diagnostic in finding the failure modes of pendulum and CIFAR10 models. An example is that when we are able to verify that the pendulum model satisfies energy conservation more - the long term dynamics of the model always reaches a stable equilibrium.

---

> ### Author Response · Authors · 2018-11-16
> **(continued)**
>
> Comment 3: The idea of generalizing verifications to a convex-relaxable set is interesting, however, applying it in general is not very clear.
>
> Answer 3: The framework outlined in the paper is general, however, for verification to be meaningful the bound is required to be sufficiently tight. We have hence approached this on a case by case basis, as getting the convex hull of an arbitrary set is hard (which is what would ensure the tightest bound). A trivial recipe could be given by a bounding box whose bounds are given by the upper and lower bounds of the sets, but in general this is not sufficiently tight. For sufficiently tight convex-relaxations, we need to make use of functional constraints which are specific to the function itself. There has been a lot of work in approximation algorithms (see http://www.designofapproxalgs.com/book.pdf for a general overview) which try to give provable guarantees by approximating this problem. The cases we have chosen to focus on, namely semantics; physics and downstream specifications, are ones we think are important, thus we have chosen to develop convex-relaxations for these specific specifications. In addition, we have included an extra example in Appendix I regarding entropy specifications please refer to Answer 1 for more details.
>
> Comment 4:  [One of my main concerns is regarding the relaxation step. There is no discussion on the effects of the tightness of the relaxation [...] especially when there is a trade-off between better approximation to the verification function and tightness of the bound.]
>
> Answer 4: In Figure 1, we have attempted to address the tightness of the relaxation. Here, we show two bounds: adversarial bound (blue line) and verification bound (green line). One thing to make clear is that there exists no verification algorithm which can have a bound past the adversarial bound. In other words, the difference between the two bounds is a strong indicator of how tight our verification algorithm is. An example is the first plot of Figure 1. Here, the difference between the adversarial bound and the verification bound is at most only 0.9% for CIFAR 10 test set. Intuitively, this means that we failed to find a provable guarantee for only 90 points out of 10000 in the test set. For the other 19910 we are able to either find an adversarial example which violates the specification or a provable guarantee that the specification is satisfied.
>
> In all cases better approximations of the verification function should give tighter bounds, we don’t expect a trade-off between the two. However, one trade-off which is important in verification is between the computational costs and the quality of the approximation of the verification function.
>
> Additionally we agree that it is desirable to understand how different relaxations can affect the tightness of the algorithm. To  address this we added Appendix H (Comparison of Tightness), where we compare two different relaxation techniques for the physics based specification (conservation of energy). In brief: we consider two different relaxations to the quadratic function, one using semi-definite programming (SDP) techniques and one using linear programming. We find that the SDP relaxation does give tighter bounds, but comes at additional computational costs.

---

> ### Author Response · Authors · 2018-11-16
> **(continued)**
>
> Comment 5: “I barely found the experimental results satisfying. To find "reasonable" inputs to the model, authors considered perturbing points in the test set. However, I am not sure if this is a reasonable assumption [...]”
>
> Answer 5: We do make the assumption that for the verification task, we should be given both a pre-trained network and a held-out set to do verification on.
>
> It’s true that in the ideal case, we would be able to verify for all possible inputs in the true distribution, however, in practice this is infeasible. Therefore, verification on a held-out set is considered a suitable proxy in the same way that accuracy on a validation/test data set is considered a suitable proxy and this has been a way to measure robustness in both verification and adversarial communities (see [Dvijotham et al., 2018; Bunel et al., 2017; Athalye et al., 2018; Carlini & Wagner, 2017b; Uesato et al. (2018); Madry et al., 2017; Tjeng & Tedrake, 2017; Cheng et al., 2017; Huang et al., 2017; Ehlers, 2017; Katz et al., 2017; Weng et al., 2018; Wong & Kolter, 2018]).  Both prior work and our experiments in Section 4.3 and 4.4 indicate that robustness on the test points is informative.
>
> Comment 6: “It is hard to have a sense of how good the results are in Figure 1 due to lack of benchmark results.”
>
> Answer 6: The reason we did not include comparisons to benchmark results from literature is that, to the best of our knowledge, this is the first paper which attempts to verify non-linear specifications as presented in our experiments.
> To resolve the lack of existing benchmarks, we have attempted to come up with strong baseline results (blue line in Figure 1) to compare with our verification results (green line in Figure 1). The strong baselines we’ve chosen are:
> Stronger adversarial attacks by having 20 random seeds as initial states for projected gradient descent.
> For the pendulum, we note that we can discretize the entire input space (as it lies on the circle). By discretizing the input space into smaller subspaces to do verification on - this is as close as we can get to the true bound. Thus we can treat the exhaustive verification (blue line) as pseudo ground truth.
>
> One thing we want to emphasize again is that since there are no baselines that can do better than the blue line (adversarial bound), the difference between the green and blue line gives us an accurate measure of how suboptimal our algorithm is.
>
> An example, to see how good the results are, is the pendulum (third picture in Figure 2). Here, we see that at perturbation radius delta=0.01, the exhaustive verification gives exactly the same bound as our verification scheme. This means that for this perturbation radius we have essentially found the true percentage of the test set which satisfies the specification. As we increase this perturbation radius to delta=0.06 we find that the difference between the verification bound and exhaustive verification bound is 22%. We had 27000 points in our test set, this means the number of points where we are unable to prove is 5940, but for the rest (21060 points) we are either able to find an adversarial attack which is successful or a proof that specification is satisfied via verification.
>
> Comment 7: [The experimental results are very limited. Suggestion to run more experiments on more data sets and re-running them with more settings. N=2 for digit sums looks limited.]
>
> Answer 7: We thank the reviewer for this comment. We extended the results for the digit sum problem as suggested. We want to also respectfully note that the number of datasets considered in this paper is in line with other papers in the space of verification and adversarial robustness [Madry et al., 2017; Athalye et al., 2018; Uesato et al. (2018);Carlini & Wagner, 2017b; ;Dvijotham et al., 2018; Bunel et al., 2017; Tjeng & Tedrake, 2017; Cheng et al., 2017; Huang et al., 2017; Ehlers, 2017; Katz et al., 2017; Weng et al., 2018; Wong & Kolter, 2018], and that the compute used to perform all experiments in this paper is already extensive (e.g. the exhaustive verification for the pendulum baseline).
> We have now added experimental results for the digit sum problem for N=3 and N=4 in Appendix H.1. In brief: As expected the verification bound becomes looser for larger N, since error accumulates when summing up more digits, however with increasing N performance stabilizes. We have also added Appendix I on what we call entropy specification which we referred to in reply to your first comment about significance (see comment on entropy specification).

---

### Official Review · AnonReviewer3 · 2018-11-02
**Interesting proposal to craft non-linear, convex relaxable specifications for more complicated networks**

**Rating:** 5
**Confidence:** 3

**Review:**

This paper considers more general non-linear verifications, which can be convexified, for neural networks, and demonstrate that the proposed methodology is capable of modeling several important properties, including the conversation law, semantic consistency, and bounding errors.

A few other comments

*) Is it critical that the non-linear verifications need to be convex relaxable. Recently, people have observed that a lot of nonconvex optimization problems also have good local solutions. Is it true that the convex relaxable condition is only required for provable algorithm? As the neural network itself is nonconvex, constraining the specification to be convex is a little awkward to me.

*) The paper contains the example specification functions derived for three specific purpose, I'm wondering how broad the proposed technique could be. Say if I need my neural network to satisfy other additional properties, is there a general recipe or guideline. If not, what's the difficulty intuitively speaking?

The paper needs to be carefully proofread, and a lot of commas are missing.

---

> ### Author Response · Authors · 2018-11-16
> **Thanks you for the review**
>
> From your comments it seems that there was a misunderstanding regarding the general applicability of our method. We have updated the paper and provided extensive additional explanations below (please also consider our reply to all authors). To address the comments you have made:
>
> Comment 1: Is it critical that the non-linear verifications need to be convex relaxable. Recently, people have observed that a lot of nonconvex optimization problems also have good local solutions. Is it true that the convex relaxable condition is only required for provable algorithm? As the neural network itself is nonconvex, constraining the specification to be convex is a little awkward to me.
>
> Answer 1:  For verification purposes it is indeed critical that we have either the global optimum value or an upper bound on the global optimum value. Verification of neural networks tries to find a proof that the specification, F(x, y) <= 0, is satisfied for all x and y within a bounded set (https://arxiv.org/abs/1803.06567). Note that this condition is equivalent to max_{x,y} F(x,y) <= 0, thus if we have the global maximum -  the problem is solved. However, to find the global optimum value is often NP-hard even for ReLU networks (https://arxiv.org/abs/1705.01320). We can try to find a lower bound to the global optimum value by doing gradient descent to maximize the value of F(x,y). This is called a falsification procedure (as explained in Section 3.1). However, even if the value found is not greater than zero this is not sufficient to give a guarantee that there exists no x and y which can violate the specification, as the value is always a lower bound to the global optimum. Thus, we are motivated to find provable upper bounds on max F(x, y), ie, a number U such that F(x, y) <= U for all x, y in the input and output domain. If this U <=0 then we have found a guarantee that the specification is never violated. In order to do this, we study convex relaxations of this problem that enable computation of provable upper bounds.
>
> We also do not require the specification to be convex (for example the physics specification isn’t if Q is not a semi-definite matrix), the specification can be some complicated nonlinear function - we just require that it be convex-relaxable, which is a weaker requirement. We slightly rephrased Section 3 to make this point more obvious.
>
> Comment 2: The paper contains the example specification functions derived for three specific purpose, I'm wondering how broad the proposed technique could be. Say if I need my neural network to satisfy other additional properties, is there a general recipe or guideline. If not, what's the difficulty intuitively speaking?
>
> Answer 2: This proposed technique is capable handling all specifications which are convex-relaxable, i.e. any specification for which the set of values that (x, y, F(x, y)) can take can be bounded by a convex set. The difficulty here is always getting a tight convex set on the specification you would like to verify for.  There is a lot of literature in finding tight convex sets (https://eng.uok.ac.ir/mfathi/Courses/Advanced%20Eng%20Math/Linear%20and%20Nonlinear%20Programming.pdf), we have chosen to demonstrate the generality of our framework with three specifications that we deem to be important. In general any convex-relaxable specification can be treated in the same manner as in the paper but, of course, finding a tight convex set should be done on a case-by-case basis. We added an additional example, going beyond quadratic constraints in Appendix I. Here we verify that a given classifier is never overly confident, in other words; we would like to verify that a threshold on the entropy of the class probabilities is never violated.
>
> We would also like to emphasize that this paper is aimed to do post-hoc verification, where we consider a scenario that we are given a pre-trained neural network. Thus this is different to training your neural network to satisfy desirable properties, it is rather a safety measure before the network is put into deployment for real world applications.
>
> Comment 3: [The reviewer also commented on a lack of commas]
> Could you please expand upon this point ?

---

> > ### Comment · AnonReviewer3 · 2018-11-28
> > **My concerns are addressed.**
> >
> > Thanks for expanding the explanation on the high level idea of this paper. To me, these high level ideas matter much more than technical derivations or extensive experimental results. I think this paper can be accepted.

---

### Official Review · AnonReviewer2 · 2018-11-06
**good paper, with minor issues**

**Rating:** 7
**Confidence:** 5

**Review:**

This paper uses convex relaxation to verify a larger class of specifications
for neural network's properties. Many previous papers use convex relaxations on
the ReLU activation function and solve a relaxed convex problem to give
verification bounds.  However, most papers consider the verification
specification simply as an affine transformation of neural network's output.
This paper extends the verification specifications to a larger family of
functions that can be efficiently relaxed.

The author demonstrates three use cases for non-linear specifications,
including verifying specifications involving label semantics, physic laws and
down-stream tasks, and show some experiments that the proposed verification
method can find non-vacuous bound for these problems. Additionally, this paper
shows some interesting experiments on the value of verification - a more
verifiable model seems to provide more interpretable results.

Overall, the proposed method seems to be a straightforward extension to
existing works like [2]. However the demonstrated applications of non-linear
specifications are indeed interesting, and the proposed method works well on
these tasks.

I have some minor questions regarding this paper:

1) For some non-linear specifications, we can convert these non-linear elements
into activation functions, and build an equivalent network for verification
such that the final verification specification becomes linear. For example, for
verifying the quadratic specification in physics we can add a "quadratic
activation function" to the network and deal with it using techniques in [1] or
[2].  The authors should distinguish the proposed technique with these existing
techniques. My understanding is that the proposed method is more general, but
the authors should better discussing more on the differences in this paper.

2) The authors should report the details on how they solve the relaxed convex
problem, and report verification time. Are there any tricks used to improve
solving time? What is the largest scale of network that the algorithm can
handle within a reasonable time?

3) The detailed network architecture (Model A, Model B) is not shown. How many
layers and neurons are there in these networks? This is important to show the
scalability of the proposed method.

4) For the Mujoco experiment, I am not sure how to interpret the delta values
in Figure 1. For CIFAR I know it is the delta of pixel values but it is not
clear about the delta in Mujoco model. What is the normal range of predicted
numbers in this model?  How does the delta compare to it? Is the delta very
small or trivial?

5) Is it possible to show how loose the convex relaxation is for a small toy
example? For example, the specification involving quadratic function is a
good candidate.

There are some small glitches in equations:

* In (4), k is undefined
* In (20), I am not sure if it is equivalent to the four inequalities after (22).
There are 4 inequalities after (22) but only 3 in (20).


Many papers uses convex relaxations for neural network verification. However
very few of them can deal with general non-linear units in neural networks.
ReLU activation is usually the only non-linear element than we can handle in
most neural network verification works. Currently the only works that can
handle other general non-linear elements are [1][2]. This paper uses more
general convex relaxations than these previous approaches, and it can handle
non-separable non-linear specifications. This is a unique contribution to this
field. I recommend accepting this paper as long as the minor issues mentioned
above can be fixed.

[1] "Efficient Neural Network Robustness Certification with General Activation
Functions" by Huan Zhang, Tsui-Wei Weng, Pin-Yu Chen, Cho-Jui Hsieh, Luca Daniel.
NIPS 2018

[2] "A dual approach to scalable verification of deep networks." by
Krishnamurthy Dvijotham, Robert Stanforth, Sven Gowal, Timothy Mann, and
Pushmeet Kohli. UAI 2018.

---

> ### Author Response · Authors · 2018-11-16
> **We thank the reviewer for the detailed feedback and encouraging review.  We address individual comments below.**
>
> Comment 1: [The authors should distinguish the proposed technique to techniques from [1] and [2] which could be used to convert some non-linear specifications to linear specifications.]
>
> Answer 1: We thank the reviewer for highlighting this point, we have now added a paragraph in the section ‘Specifications Beyond Robustness’ to distinguish between existing techniques and convex relaxable specifications. The reviewer is correct in pointing out that some non-linearities can indeed be linearized through the use of different element-wise activation functions. However, in terms of generality as the reviewer mentioned, this mechanism does not work in many cases - an example is the softmax function, which needs every input in the layer to give it’s output. In this particular case, it is a non-separable nonlinear function and current literature does not support verification with such non-linearities.

---

> > ### Comment · AnonReviewer2 · 2018-11-26
> > **Thanks for the updates. I am okay with this paper.**
> >
> > Dear Paper915 Authors,
> >
> > Thanks for clarifying my concerns and adding new materials on the toy example and scalability to the paper. I am okay with this paper now if the AC wants to accept it.
> >
> > BTW please make sure also adding detailed structures of each model evaluated to the appendix, or release source code with model specifications.
> >
> > Thanks,
> > Paper915 AnonReviewer2

---

> ### Author Response · Authors · 2018-11-16
> **(Continued)**
>
> Comment 2: “Report the details on how they solve the relaxed convex problem, and report verification time. What is the largest scale of network that the algorithm can handle within a reasonable time?”
>
> Answer 2: Thanks for the suggestion. We have added Appendix E (Scaling and Implementation) where we explain how we solved the relaxed convex problem. For CIFAR 10 semantic specification and downstream task specification (since all constraints are linear) we have used the open source LP solver GLOP (https://developers.google.com/optimization/lp/glop) and on average this takes 3-10 seconds per data point on a desktop machine (with 1 GPU and 8G of memory) for the largest network we handled. This network consists of 4 convolutional and 3 linear layers comprising of in total 860000 parameters. For the conservation of energy, we used SDP constraints - this relaxation scales quadratically with respect to the input and output dimension of the network. To solve for these set of constraintt we used the CVXOPT solver (https://cvxopt.org/) accessed via the python interface CVXPY (http://www.cvxpy.org/), which is slower than GLOP, and we have only tested this on a small network consisting of two linear layers with a total of 120 parameters.  However, we expect that with stronger SDP solvers (like Mosek - https://www.mosek.com/) or by using custom scalable implementations of SDPs (for example, the techniques described in https://people.eecs.berkeley.edu/~stephentu/writeups/first-order-sdp.pdf), we will be able to scale to larger problem instances - we plan to pursue this in future work.

---

> ### Author Response · Authors · 2018-11-16
> **(Continued)**
>
> Comment 3: [Detailed network architecture (Model A, Model B).  Comment on the scalability of the proposed method]
>
> Answer 3: For the CIFAR 10 Semantic Specification, Model A and Model B are identical in terms of network architecture and consist of 4 convolutional and 3 linear layers interleaved with ReLU functions and 860000 parameters. For the MNIST Downstream Task the models consist of two linear layers interleaved with ReLU activation and 15880 parameters - which was enough to get good adversarial accuracy. For the pendulum physics specification we used a two layer neural network with ReLU activations and in toal 120 parameters. Regarding the scalability please see our previous comment.

---

> ### Author Response · Authors · 2018-11-16
> **(Continued)**
>
> Comment 4: [For the Mujoco experiment, I am not sure how to interpret the delta values in Figure 1. Is the delta trivial?]
>
> Answer 4: Thanks pointing this out, we should have been clearer about this. We have added these details to the appendix. For completeness we also list them here.
> - The pendulum model takes [cos(theta), sin(theta), v] as input. Here theta is the angle of the pendulum and v is the scaled angular velocity (angular velocity / 10) - the data is generated such that the initial angular velocity lies between (-10, 10), by scaling this with 10 we make sure [cos(theta), sin(theta), v] lies in a box where each side is bounded by [-1, 1].
> - The pendulum setup is the Pendulum from the DeepMind Control Suite (https://arxiv.org/abs/1801.00690). The pendulum is of length 0.5m and hangs 0.6m above ground.
> - When the perturbation radius is 0.01. Since the pendulum is of length 0.5m, the perturbation in space is about 0.005 m in the x and y direction. The perturbation of the angular velocity is true_angular_velocity +- 0.1 radians per second (since the input is a scaled angular velocity by a factor of 10). The largest delta value we verified for was 0.06, this means the angular velocity can change upto 0.6 radians per second which is about ⅕ of a full circle, thus this is not a trivial perturbation.

---

> ### Author Response · Authors · 2018-11-16
> **(Continued)**
>
> Comment 5: “Is it possible to show how loose the convex relaxation is for a small toy example? For example, the specification involving quadratic function is a good candidate.”
>
> Answer 5: We have now added a section in the Appendix H.2 (Tightness with a Toy Example), here we consider a toy example where the specification is :F(x,y) = x^2 - y^2 - 4<=0. The variables x and y are from two interval sets (-b_x, b_x) and (-b_y, b_y) respectively. Throughout the toy example we keep b_y=9. In Appendix H.2, we have added Figure 8, where the plot on the left shows the true set which satisfies the specification and we also show our convex relaxed set using our relaxation. The convex relaxed set is simply a box around the true set which is bounded hyperbolically (shown in green). In the same figure with the plot on the right, we also show the tightness of our relaxation as the interval set increase in length, specifically as we increase b_x.  What we find is that our relaxation becomes looser linearly with respect to the increase in interval length.
>
> Minor Comments:
> [In (4), k is undefined]
> Thanks for spotting this, k was indeed a typo, this is now changed to n.
>
> [In (20), I am not sure if it is equivalent to the four inequalities after (22). There are 4 inequalities after (22) but only 3 in (20). ]
> There were only three constraints in equation 20 as we enforce X- aa^T to by a symmetric semi-definite matrix. The constraints X_ij - l_j a_i - u_i a_j + l_j u_i >=0,  X_ij - l_i a_j - u_ja_i + l_i u_j >=0 becomes the same constraint when X_ij is symmetric. Thanks for spotting this, we have made this clearer in the appendix. In fact, this allowed us to spot that we also missed some constraints which we enforced this is now also added.

---

### Author Response · Authors · 2018-11-16
**We have updated the paper and thanks for all the reviews.**

We thank all reviewers for the detailed reviews and thoughtful remarks. We have addressed all concerns in an updated version of the paper and you can find responses to your questions below.

We would like to clarify two points that came up in multiple reviews:
1) The nonlinear specifications that can be verified with our method do not have to be convex. We only require the specification to convex-relaxable - which is a weaker condition. We have rephrased parts of Section 3 to make this more clear.
2) The framework outlined in the paper is general, however, for verification to be meaningful the bound is required to be sufficiently tight, which requires a tight convex-relaxation that is dependent on the form of the function and thus has to be problem specific. We also refer to Answer 3 to Reviewer 1 for a more detailed response.

---

### Meta-Review · Area_Chair1 · 2018-12-17
**Interesting contribution to understanding NNs**

**Confidence:** 5
**Recommendation:** Accept (Poster)

**Metareview:**

This paper proposes verification algorithms for a class of convex-relaxable specifications to evaluate the robustness of neural networks under adversarial examples.

The reviewers were unanimous in their vote to accept the paper. Note: the remaining score of 5 belongs to a reviewer who agreed to acceptance in the discussion.